# Kiwifruit Vine Decline Syndrome (KVDS) Alters Soil Enzyme Activity and Microbial Community

**DOI:** 10.3390/microorganisms12112347

**Published:** 2024-11-16

**Authors:** Valentino Bergamaschi, Alfonso Vera, Lucia Pirone, José A. Siles, Rubén López-Mondéjar, Laura Luongo, Salvatore Vitale, Massimo Reverberi, Alessandro Infantino, Felipe Bastida

**Affiliations:** 1Department of Environmental Biology, Sapienza University of Rome, 00185 Rome, Italy; massimo.reverberi@uniroma1.it; 2Council for Agricultural Research and Economics (CREA), Research Centre for Plant Protection and Certification (CREA-DC), 00156 Rome, Italy; lucia.pirone@crea.gov.it (L.P.); laura.luongo@crea.gov.it (L.L.); salvatore.vitale@crea.gov.it (S.V.); alessandro.infantino@crea.gov.it (A.I.); 3Department of Soil and Water Conservation and Organic Waste Management, CEBAS-CSIC, 30100 Murcia, Spain; avera@cebas.csic.es (A.V.); jsiles@cebas.csic.es (J.A.S.); rlmondejar@cebas.csic.es (R.L.-M.); fbastida@ceabas.csic.es (F.B.)

**Keywords:** KVDS, microbial community, soil enzymatic activity, PLFAs

## Abstract

Kiwifruit Vine Decline Syndrome (KVDS) has become a major concern in Italy, impacting both plant health and production. This study aims to investigate how KVDS affects soil health indicators and the composition of soil microbial communities by comparing symptomatic and asymptomatic areas in two kiwifruit orchards located in Latium, Italy. Soil samples were collected during both spring and autumn to assess seasonal variations in soil physicochemical properties, enzyme activities, and microbial biomass. The results reveal that KVDS influences several soil properties, including pH, electrical conductivity, and the contents of water-soluble carbon and nitrogen. However, these effects varied between orchards and across different seasons. Additionally, KVDS significantly impacts soil enzyme activities and microbial biomass, as assessed through the phospholipid fatty acid (PLFA) analysis, particularly showing an increase in fungal biomass in symptomatic areas. Metabarcoding further demonstrates that microbial communities differ between symptomatic and asymptomatic soils, exhibiting notable shifts in both diversity and relative abundance. Our findings emphasise the complex interactions between plants, soil, and microbial communities in relation to KVDS. This suggests that the syndrome is multifactorial and likely linked to an imbalance in soil microbial communities at the rhizosphere level, which can negatively affect soil health.

## 1. Introduction

The kiwifruit [(*Actinidia deliciosa* (A. Chev.) C. F. Liang et A. R. Ferguson var. *deliciosa*)] is a species of considerable agricultural, botanical, and economic importance. Native to China, it is now grown globally because of its high nutritional content, excellent flavour, and health benefits [1]. With an estimated value of 3 billion dollars worldwide, kiwifruit production has a relevant economic impact on the primary industry of the main producers. Italy is the third global producer, with a production of 523,120 tonnes harvested from 24,850 productive hectares [2]. Half of Italian production is concentrated in the Latium region, with an area of 8888 ha and 250,252 tons of fruit harvested in 2023 [3]. As for many other fresh fruits and vegetables, the kiwifruit crop is affected by several pests and diseases that can potentially pose a risk to production sustainability. Due to its root anatomy and development [4], kiwifruit is particularly susceptible to high temperatures [5] and anaerobic conditions, mainly waterlogging [6,7]. Early reports of damages caused by oomycetes proliferating in excess water conditions have been documented [8,9,10,11]. Very recently, a new kiwifruit disorder, the Kiwifruit Vine Decline Syndrome (KVDS), has been reported almost exclusively in Italy. First reported in some orchards in the Veneto region in 2012 [12], it has quickly become a leading concern in several kiwifruit-growing areas along the peninsula, causing significant yield reduction and even crop mortality [13]. To the best of our knowledge, the disease remains confined to Italy, though similar vine decline disorders likely attributable to KVDS and linked to damage caused by oomycetes thriving in waterlogged conditions [8,9,10,11] have been recently reported in Turkey [14,15,16,17] and China [18].

The symptomatology of KVDS is primarily evident in the root system of the affected plants, showing widespread browning and large rotten portions with the presence of typical rat tails. The lack of absorbent roots leads to the loss of root functions, which can then impair the epigeal part of the plant, resulting in centripetal leaves drying, leaf drop, fruit fall, and ultimately plant death [19]. Several oomycetes (e.g., *Phytophthora* spp., *Pythium* spp., and *Phytopythium* spp.), fungi (e.g., *Cylindrocarpon* spp., *Fusarium* spp., and *Pyrenochaeta* spp.), and bacteria (*Clostridium* spp., *Erwinia* spp.) or their secondary metabolites produced have often been associated with or isolated from KVDS affected roots, but their role as the leading cause of the disease has not yet been clearly demonstrated [20,21].

There is an intimate interaction between crops and soil microbial communities [22,23] that influences soil health. Thus, crops influence soil microbial communities’ composition, diversity, and functioning. Soil microbiota, including bacteria, fungi, archaea, and other microorganisms, play a key role in soil functions, providing plant nutrients, forming soil aggregates, decomposing organic matter, suppressing diseases, and mitigating climate change [24,25,26,27]. Soil microbial community diversity and abundance can lead to changes in nutrient cycling and soil organic matter dynamics [28]. In order to track the soil’s biological health, we have used a suite of several indicators. For instance, the activity of the soil microbial community was characterised by measuring soil respiration and extracellular enzyme activities associated with carbon (C), nitrogen (N), and phosphorus (P) cycling [29], and microbial biomass was assessed through the phospholipid fatty acid (PLFA) analysis [30]. Bacterial and fungal community diversity and composition were analysed by sequencing the 16S rRNA gene and internal transcribed spacer (ITS) regions, respectively. Here, we aim to evaluate the impact of KVDS on soil health indicators and microbial communities by comparing symptomatic and asymptomatic kiwifruit plants and their associated microbiota. Considering the strong interaction between plant and soil microbial communities in the rhizosphere [22,23] and the fact that KVDS is a disorder with strong disruption of the root system, we hypothesised that KVDS can affect the enzymatic activity of rhizosphere soil, with changes in the composition and diversity of soil microbial communities exhibiting seasonal differences, including variations in biomass, activity, and composition, due to changes in environmental conditions and disease progression.

## 2. Materials and Methods

### 2.1. Experimental Design and Soil Sampling

This study was conducted in two orchards located in Latium, Italy. Both orchards were subjected to integrated management with similar farming techniques. Orchard 1 (O1) was in Lanuvio (south of the province of Rome, 41°37′14.2″ N 12°43′06.0″ E, O1S—41°37′11.8″ N 12°43′09.8″ E) and Orchard 2 (O2) in Aprilia (province of Latina, 41°33′33.9″ N 12°43′45.2”E; O2S—41°33′31.1″ N 12°43′49.7″ E). Soil from Orchard 1 was classified as Haplic Phaeozems (25–50%), Luvic Phaeozems (10–25%), and Cambic Endoleptic Phaeozems (<10%). The surrounding landform was described as slopes and eroded plateau surfaces on pyroclastic products, mainly consolidated (tuffs). Those from Orchard 2 were classified as Haplic Luvisols (25–50%) and Protovertic Endogleyic Cambisols (25–50%). The surrounding landform was described as “high” Pontine plain surfaces on prevailing fluvial deposits [31]. In each orchard, both asymptomatic (A) and symptomatic (S) areas were identified (O1A and O1S for Orchard 1, O2A and O2S for Orchard 2). Four plants were randomly selected (replicates, *n* = 4) within each symptomatic and asymptomatic area of each orchard as healthy controls (Figure 1a,b). For each plant, five sub-samples of rhizosphere soil were collected around the trunk, about 50 cm far from the collar (Figure 1c,d), to create a composite soil sample (ca. 3 kg). Soil sampling was conducted twice a year (spring and autumn) to evaluate the possible influence of phenology and crop season. Soil samples were air-dried at room temperature in the lab and sieved to <2 mm before storage at 4 °C for chemical and biochemical analyses and at −20 °C for phospholipid fatty acid (PLFA) and molecular analyses.

### 2.2. Disease Severity Score 

A scoring scale from 0 to 3 for disease severity for both epigeal and hypogeal symptoms was set up to better distinguish between symptomatic and asymptomatic plants and accurately categorise them into different classes. For epigeal symptoms, 0 = no symptoms (healthy plant); 1 = mild symptoms (plant decay sometimes identifiable by declining new shoots and the appearance of leaf chlorosis, few new shoots, reduced leaf size, widespread leaf chlorosis); 2 = severe symptoms (phylloptosis, carpoptosis, and reduced fruit size, (if present), vine decline); and 3 = dead plant. For hypogeal symptoms, 0 = no symptoms (healthy roots); 1 = mild symptoms (reduced adsorbent root and widespread necrosis); 2 = severe symptoms (some primary roots and almost all secondary roots rot and loss of cortical tissues begins, showing the rat’s tails and almost disappearing absorbent roots); and 3 = dead plant (Figure 2). Using the described disease severity scale, it was possible to select the asymptomatic areas to collect soil samples from the rhizosphere of healthy kiwifruit trees.

### 2.3. Soil Physicochemical Properties

The water content of the soil was measured gravimetrically. Soil pH, electrical conductivity (EC), the water-soluble content of carbon (WSC), and the water-soluble nitrogen content (WSN) were measured in a soil/distilled water extract (1:5, *w:v*). Soil pH was analysed using a pH meter (9615S–10D; HORIBA Scientific, Piscataway, NJ, USA). EC was analysed using a conductivity cell (9382–10D, HORIBA Scientific, Piscataway, NJ, USA). WSC and WSN were analysed using a C/N analyser for liquid samples (Multi N/C 3100, Analytic, Jena, Germany). Ammonium (NH_4_^+^) was analysed by colourimetric determination in KCl extracts following the method by Kandeler and Gerber [32]. Soil macro- and micro-nutrient contents were measured using an ICP-OES spectrometer (ICAP 6500 DUO from Thermo-Scientific, Waltham, MA, USA) after a process of nitric-perchloric acid digestion. Total carbon (TC), total nitrogen (TN), and soil organic carbon (SOC) content were analysed using an Elemental Analyser (C/N Flash EA 112 Series-Leco Truspec, St. Joseph, MI, USA).

### 2.4. Soil Microbial Respiration and Enzyme Activities

Basal soil respiration (BSR) (mg CO_2_-C kg^−1^ soil day^−1^) was measured by placing 20 g of soil at 50% of its water holding capacity (WHC) in hermetically sealed flasks and incubating for 40 days at 28 °C in the darkness. Released CO_2_ was measured periodically with an infrared CO_2_ analyser (CheckMate 3O2 (Zr) CO_2_–100%; MOCON Europe A/S, Ringsted, Denmark). Beta-glucosidase and alkaline phosphatase activities were analysed using the methods of Eivazi and Tabatabai [33] and Tabatabai and Bremner [34], respectively. Both activities were expressed in units of micromoles of p-nitrophenol (PNP) produced per gram of dry soil per hour (μmol PNP g^−1^ h^−1^). Urease activity was determined using the buffered method of Kandeler and Gerber [32] and was expressed in units of micromoles of ammonium-N produced per gram of dry soil per hour (μmol NH_4_^+^-N g^−1^ h^−1^).

### 2.5. Phospholipid Fatty Acid (PLFA) Analysis

Soil microbial biomass was determined by phospholipid fatty acid analysis (PLFA). Fatty acids were extracted from soil using a chloroform/methanol/citrate buffer (1:2:0.8 *v/v/v*) according to Bligh and Dyer [35]. Fatty acids were then fractionated to obtain the phospholipidic fraction [30] and then transformed into fatty acid methyl esters [36]. The PLFAs were analysed with a gas chromatograph (8860 GC System, Agilent Technologies, Santa Clara, CA, USA) equipped with a flame ionisation detector (FID), using a DB–Fast FAME capillary column (30 m × 0.25 mm ID × 0.25 μm film) (Agilent Technologies), with helium as the carrier gas. The conditions used were as follows: an initial temperature of 80 °C for 1 min 30 s, then an increase to 160 °C with a ramp of 40 °C/min, then to 167 °C at 0.5 °C/min, then to 200 °C at 30 °C/min, and finally to 230 °C at 4 °C/min. All fatty acids mentioned in this article are described according to the standard nomenclature of Vestal and White [37]. The fatty acids i15:0, a15:0, i16:0, i17:0, 16:1ω7, cy17:0, cy19:0, 10Me16:0, and 10Me18:0 are considered representative of bacterial biomass [30,38], and the fatty acids 18:2ω6,9t and 18:2ω6,9c are considered representative of fungal biomass [39,40]. The fatty acids i15:0, a15:0, i16:0, i17:0, 10Me16:0, and 10Me18:0 are considered representative of Gram-positive (Gram+) biomass, while fatty acids 16:1ω7, cy17:0, and cy19:0 are considered representative of Gram-negative (Gram−) biomass [30,38]. The actinobacterial representative fatty acids are 10Me16:0 and 10Me18:0 [38].

### 2.6. DNA Extraction

Extraction of DNA from 250 mg of soil from each sample was performed using a DNeasy PowerSoil Pro Kit (QIAGEN, Hilden, Germany) following the manufacturer’s instructions with a modified homogenisation step using the FastPrep Instrument (MP Biomedicals, Santa Ana, CA, USA) instead of vortexing. DNA samples were quantified and quality checked using the DeNovix dsDNA High Sensitivity Assay kit (DeNovix) and the DeNovix DS-11 FX+ Spectrophotometer/Fluorometer (DeNovix Inc., Wilmington, DE, USA), respectively. DNA concentration for each extraction was standardised to 20 ng/μL^−1^, and only 3 low-concentrated samples were standardised at 5 ng/μL^−1^.

### 2.7. Bacterial V4 and Fungal ITS2 Amplicon Sequencing

The V4 region of the bacterial 16S rRNA gene was amplified using barcoded primers 515F and 806R [41]. The PCR amplification of DNA from the fungal ITS2 region was performed using barcoded primers gITS7 and ITS4 [42] in three PCR reactions per sample, as described by Žifčáková, et al. [43]. Both primers were composed of the barcode (4–6 nucleotides), a spacer (2 nucleotides) absent in all GenBank sequences at this position to avoid preferential amplification of some targets [44], and the specific primer. Both forward and reverse primers were barcoded to make sure that barcode-switching did not affect the results and to avoid problems with Caporaso primers [41]. The PCR products were purified using a MiniElute Kit (Qiagen), and the concentrations were measured with a Qubit 4 Fluorometer (ThermoFisher Scientific). Afterwards, the library was prepared according to Vera et al. (2021) [45], and sequencing of microbial amplicons was performed on Illumina MiSeq (in a paired-end 2 × 300 base pair (bp) run). Bioinformatic processing of the sequences was conducted using the USEARCH pipeline and UPARSE-OTU algorithm [46]. The paired-end (PE) sequences were first merged with the command -fastq_mergepairs. Then, PE reads were quality-filtered, allowing a maximum e-value of 1.0, trimmed (to 240 and 250 bp for prokaryotic and fungal libraries, respectively), dereplicated, and sorted by abundance (removing singletons) prior to chimaera detection and determination of OTUs (operational taxonomic units) at 97% sequence identity. Finally, the original sequences were mapped to OTUs at the 97% identity threshold to obtain one OTU table for the prokaryotic community and one OTU table for the fungal community. The taxonomic affiliation of each OTU was obtained using the -syntax algorithm against the RDP 16S rRNA training set for 16s rRNA gene sequences [47] and UNITE for ITS2 sequences [48], with an 80% confidence threshold in both cases. The sequencing depth across libraries was normalised, and the normalised OTU tables were used for downstream analyses. Diversity indicators (Shannon–Wiener index -H- and richness -R-) were calculated for the bacterial community (16S rRNA sequences) and the fungal community (ITS sequences) using the command -alpha_div. The DNA sequences have been submitted to the NCBI SRA with the assigned accession number PRJNA1156594.

### 2.8. Statistical Analysis

The data normality and homoscedasticity were checked with the Kolmogorov–Smirnov and Levene tests, respectively. To assess the effect of the factor “plant’s status”, data, including diversity indicators (Shannon–Wiener index -H- and richness -R-) from each orchard at each sampling time, were subjected to a one-way ANOVA. The data were further subjected to a two-way repeated measures analysis of variance (ANOVA) to test the interactive effects of the “plant’s status” (Symptomatic, Asymptomatic) and “orchard” (O1, O2) within the same sampling period. The effects of the factor “plant’s status” and its interaction with the other independent factors on seasonal variations were tested in a three-way repeated measures ANOVA. The ANOVA test was followed by post hoc Tukey’s significant difference test. Differences at *p* ≤ 0.05 were considered statistically significant.

To assess variations in soil microbial structure (bacterial and fungal communities) between asymptomatic and symptomatic areas in each of the two orchards, a non-metric multidimensional scaling (NMDS) analysis was conducted using Bray–Curtis dissimilarities at the OTU level. The significance of the differences was assessed via PerMANOVA with 9999 permutations using the adonis2 function in R version 4.2.1 with the “vegan” package. Indicator Species Analysis (ISA) was used to link relative abundance at the OTU level with orchard status running the “indicspecies” package in R version 4.2.1. A threshold level of indicator value with 95% significance (*p*-value ≤ 0.05) was chosen as a cut-off for identifying indicator species [49].

## 3. Results

### 3.1. Disease Severity Scale 

A disease assessment scale has been used to better characterise the plant disease status in all studied areas and to correlate the results of the soil parameters with the severity of KVDS symptoms. The severity of KVDS in the symptomatic and asymptomatic areas of both orchards (O1 and O2) in the two seasons is reported in Table 1. The scoring scale of the epigeal part was used to define the correct asymptomatic and symptomatic status in both orchards and seasons. Despite the absence of epigeal symptoms, the hypogeal scoring indicated an initial disease symptom in asymptomatic plants in Orchard 2 from spring onwards.

### 3.2. Soil Physicochemical Properties

The results of the One-Way ANOVA for pH, EC, WSC, and WSN are shown in Figure 3 and detailed in Table 2. Among them, significantly higher pH values were only found in O1 symptomatic soils in both seasons compared to asymptomatic ones. Higher EC values were observed in O1 symptomatic soils in both seasons (*p* < 0.05), whereas they were lower in O2 symptomatic soils only in spring. In spring, no significant differences in WSC were recorded between symptomatic and asymptomatic soils for both O1 and O2 soil samples. However, in autumn, WSC was lower in the soil of O1 from symptomatic plants, whereas it was higher in O2 plants (*p* < 0.05). WSN values were significantly different only in spring, higher for O1 symptomatic plants and lower for O2 symptomatic plants. Two-way ANOVA results of the spring sampling (Appendix A) showed significant O × S interactions for pH, EC, and WSN, whereas in autumn, this interaction was significant only for pH and WSC. The triple O × S × t interaction of the measured parameters was significant for all parameters except pH (Appendix A). Significantly higher values of SOC and TN were recorded in spring for O1 and in autumn for O2.

### 3.3. Soil Basal Respiration and Enzyme Activities

The enzyme activities showed overall distinct patterns (Table 2). In both seasons, the β-glucosidase activity was significantly higher in O1 symptomatic soils than in the corresponding asymptomatic areas (Figure 4a). Regarding alkaline phosphatase, the soils in the symptomatic areas in spring (O1) showed significantly higher activity than those in the corresponding asymptomatic areas (Figure 4b).

Urease activity was slightly higher in symptomatic soils collected in O2 in autumn (Figure 4c) than in asymptomatic soils. With the only exception of O1 in autumn, a general trend of significantly increased basal respiration of the symptomatic soils was observed (Figure 4d). Two-way ANOVA O × S interaction was highly significant for β-glucosidase and BSR in spring. In contrast, it was significant for Urease and BSR in autumn. Three-way ANOVA showed significant O × S × t interactions for all four parameters measured (Appendix A).

### 3.4. Phospholipid Fatty Acid (PLFA) Analysis

The microbial biomass of the soil was assessed by the soil PLFA extraction, and the results are shown in Figure 5 and detailed in Table 2. As a general trend, in spring, the O1 symptomatic soils always showed significantly higher total biomass, fungi, bacteria, and fungi/bacteria ratio, whereas in O2, it was only the case for the bacterial biomass and the fungi/bacteria ratio. Conversely, in autumn, only the fungal and bacterial biomass were significantly higher in O2 symptomatic soils (Figure 5a). The O × S interaction was significant for bacteria in both spring and autumn (Appendix A), likewise to the triple O × S × t interaction. The latter was also significant for fungi (Appendix A).

The analysis of the correlation between all the measured parameters is shown in Figure 4. The positive correlations between fungi and bacteria are worth mentioning, those of β-glucosidase activity with several physiochemical parameters (WSN, TC, TN, SOC) and some enzymatic activities (alkaline P and BSR). Very interesting positive correlations have been observed that BSR positive correlates with WSC, WSN and alkaline P, while bacteria were highly correlated with N total and SOC. In addition, positive correlations were found with soil respiration and the β-glucosidase and phosphatase activities, but no significant correlation with urease. In contrast, no significant correlation was found between fungal biomass and these soil parameters (Figure 6).

### 3.5. Fungal ITS and Bacterial 16Samplicon Sequencing

To compare the microbial communities of asymptomatic and symptomatic rhizosphere soil samples, we performed amplicon sequencing of fungal ITS (ITS 2) and bacterial 16S (V4) regions. There were no significant differences in fungal diversity (Shannon and richness) across seasons and treatments (Table 3). With regard to bacterial diversity, it was higher in O1S and O2S than in the corresponding asymptomatic plants in spring, but there were no significant differences in both orchards in autumn. The relative abundance of soil microbiota at the *phylum* level for both orchards in the two sampling seasons is reported in Figure 7. As for fungi, *Ascomycota* was the most abundant phylum, followed by *Basidiomycota*, *Rozellomycota,* and *Chytridiomycota*. One-way ANOVA of the most abundant *phylum* showed differences between the two symptomatologic areas only in Orchard 1 in both sampling times (Appendix A). In both seasons, the relative abundance of *Ascomycota* was significantly higher in the soils of symptomatic than in asymptomatic trees in O1, and *Basidiomycota* was higher in the asymptomatic areas of the same orchard (Appendix A; *p* ≤ 0.05). Among the bacterial phyla, *Proteobacteria*, *Actinobacteria*, *Firmicutes*, and *Acidobacteria* were the most prevalent. Their relative abundance varied significantly with the health status of trees in Orchard 1, particularly in spring, and all except *Proteobacteria* showed similar trends in autumn. As in the case of fungi, the most noticeable differences between the bacterial phyla were observed in Orchard 1. The impact of tree health on bacterial abundance shifted over time, with significant changes in *Proteobacteria* and *Acidobacteria* during spring (Appendix A; *p* ≤ 0.001) and in *Actinobacteria* and *Planctomycetes* during autumn (Appendix A; *p* ≤ 0.001), with the symptomatic areas showing the highest relative abundances.

Since the disease is unlikely to have a bacterial origin, an Indicator Species Analysis (ISA) was conducted on the fungal community to determine which OTUs were most associated with asymptomatic and symptomatic soil samples. Sixteen and twenty-three OTUs, mostly Ascomycota, were significantly associated with the soils of the symptomatic and asymptomatic areas, respectively, and were identified as potential indicator species. In more detail, OTUs to the family *Pyronemataceae* and *Aspergillaceae* were enriched in the soils of the asymptomatic areas, whilst fungal taxa assigned to the class *Leotiomycetes* and the family *Periconiaceae* were found to be the most statistically associated with the soils of the asymptomatic areas (Table 4 and Table 5).

The structure of soil microbial communities was assessed through a non-metric multidimensional scaling (NMDS) of bacterial and fungal OTUs in both orchards (O1 and O2). The NMDS analysis showed significant differences in fungal and bacterial community composition between orchards and disease status. The PerMANOVA test confirmed the differences in NMDS, indicating significant statistical differences, as reported in Figure 8.

## 4. Discussion

### 4.1. Severity Scale Disease

Since this study aimed to evaluate the impact of Kiwifruit Vine Decline Syndrome (KVDS) on soil health indicators and microbial communities, it is of primary importance to identify areas with and without KVDS by comparing symptomatic and asymptomatic kiwifruit trees in different orchards. In fact, the kiwifruit trees affected by KVDS may appear asymptomatic above ground, while the same trees may already exhibit root symptoms. This pattern can be attributed to the disease progression [50,51,52]; mild symptoms typically emerge in early spring, and if the syndrome persists, plants may deteriorate rapidly in summer or autumn, coinciding with fruit ripening when nutrient demand peaks, explaining the worsening of root symptoms in autumn. An easy-to-use severity scale, as described in Section 2.2, helps plan early interventions and mitigate the impacts of the disease [53].

### 4.2. Impact of KVDS on Soil Physicochemical Properties and Enzymatic Activities

The results of the comparative analysis indicated that orchard characteristics, symptomatologic status, and seasonal variations significantly influenced soil properties and enzyme activities, which are critical factors in understanding and managing KVDS. In Orchard 1 (O1), soil pH was higher in soils from symptomatic trees during both seasons, while Orchard 2 (O2) showed no significant differences in soil pH based on symptomatology. Other studies have found that soil pH variations, in particular pH > 7, significantly affect nutrient solubility and microbial activity, influencing plant health and disease resistance [54,55]. However, the higher pH level observed exclusively in symptomatic areas of Orchard 1 indicates that KVDS severity cannot be directly attributed to the pH levels alone.

Enzyme activities have been widely used as indicators of soil health and quality [29,56]. Our findings indicate that enzyme activities and basal soil respiration are more strongly influenced by orchard type and seasonal variations than by the symptomatologic status of trees. Bastida et al. (2008) [56] reported that higher enzyme activities are generally associated with healthier soils and better plant growth and, depending on local soil conditions, also with management practices. Conversely, our results showed that although a general pattern cannot be concluded, some enzyme activities and basal respiration in symptomatic areas were higher than in asymptomatic areas. The explanation of these results might reflect the activity of the pathogenic microbial community in KVDS symptomatic areas as well as the existence of root rests in diseased plants that can generate substrate for more significant microbial activity. Indeed, we found a link between β-glucosidase activity and root symptom severity, suggesting that this enzyme might be secreted by phytopathogenic Oomycetes and/or other fungi to trigger the roots of diseased plants. In Orchard 2, during the spring, mild root symptoms in asymptomatic areas suggested a potential link with the β-glucosidase activity dynamics. Previous studies have reported some Oomycetes species as pathogenic to kiwifruit [15,57], and other works have noted a high β-glucosidase activity in oomycetes [58] and fungi [59]. In the same way, the higher BSR levels in symptomatic areas could be due to higher organic matter availability, as indicated by soil organic carbon levels and increased fungal biomass in soils with diseased plants, as outlined below.

### 4.3. Impacts of KVDS on Soil Microbial Community

The symptomatic status of trees and the season influenced the soil microbial biomass, with increased microbial abundance, particularly fungal biomass, in both orchards in spring, when the KVDS symptoms were present. Similar results were obtained in other studies investigating different crops, indicating an increased fungal and bacterial abundance in diseased roots [60,61]. These results highlight that KVDS can impact the microbial biomass, which is crucial for soil health [62], by influencing plant growth, nutrient cycling, and disease resistance [63] through beneficial interactions with plant roots. Although it is challenging to identify a specific pathogen associated with the disease through metabarcoding, the distinct microbial groups found in asymptomatic rhizosphere soils highlight the complex plant–microbe interactions crucial to soil health and fertility [64].

Metabarcoding confirmed differences in community composition between asymptomatic and symptomatic areas consistent with our hypothesis. Our results suggest an intense interaction between the season and plant health status in shaping fungal diversity. In spring, there was a consistent trend toward higher richness and Shannon index in asymptomatic O2 than in symptomatic O2, whereas in autumn, this distinction was less pronounced. In contrast, bacterial diversity did not exhibit a consistent trend across seasons and health states, suggesting the involvement of factors additional to seasonality and KVDS symptoms in modulating the bacterial diversity.

Further, the Indicators Species Analysis at the OTU level confirmed that *Ascomycota* was more representative in symptomatic areas and underscored the importance of groups like *Glomeromycota* and *Basidiomycota* in asymptomatic rhizosphere soil, known for positive plant interactions [65]. When beneficial microorganisms dominate, they can suppress the growth and activity of pathogens, while saprophytes continue to recycle nutrients efficiently. An imbalance, such as an overabundance of pathogens, such as *Phytopythium vexans* known to cause root rots [66,67], or a depletion of beneficial microbes, can lead to poor plant health, reduced crop yields, and increased susceptibility to diseases. It has been reported that some fungal species could produce phytotoxic exudates, such as *Dactylonectria* spp. [68], which could be involved in KVDS symptoms [12,69]. Here, we found an increase in the relative abundance of pathogenic fungi, like *Ilyonectria* spp., reported as a causal agent of necrotic lesions on woody roots [70]. In the context of KVDS, disrupting beneficial microbial populations, including those involved in nutrient cycling and pathogen suppression, exacerbates KVDS symptoms [71]. One factor contributing to the negative impact of these valuable microbes is frequent irrigation over the years, leading to progressive soil compaction along the planted rows in kiwifruit orchards. Moreover, this study allows us to relate KVDS to changes in the bacterial community. Our findings report changes in the abundance of important soil functional genera, such as Firmicutes, which includes the *Bacillus* genera, which are less represented in symptomatic areas of kiwifruit orchards. Other works suggest that microorganisms, such as *Proteobacteria*, *Firmicutes*, and mycorrhizal fungi, can alleviate abiotic stresses, such as drought, salinity, and temperature extremes, thereby enhancing plant resilience [72]. Conversely, pathogenic bacteria can significantly reduce crop yields, leading to dysbiosis, in which pathogenic microbes dominate, out-competing and suppressing beneficial organisms.

## 5. Conclusions

This study highlights the seasonal dynamics of soil physicochemical properties, enzyme activities, and microbial biodiversity in kiwifruit orchards affected by Kiwifruit Vine Decline Syndrome (KVDS). Significant differences in enzymatic activities and microbial biomass are associated with KVDS symptoms. Metabarcoding revealed distinct microbial communities in symptomatic versus asymptomatic areas. In particular, β-glucosidase activity in symptomatic rhizospheres merits further investigation as a potential indicator of plant–soil interactions. To mitigate KVDS symptoms, farmers are advised to avoid excessive irrigation, as overwatering can create waterlogged conditions that disrupt soil health and promote pathogenic microbial growth; thus, implementing balanced water management is essential. In addition, adopting conservative soil management practices is recommended to help preserve soil structure. The complex interplay between KVDS, soil health, and microbial communities highlights the need for integrated management strategies to mitigate the impact of the disease on kiwifruit crops. Further investigation will shed light on the aetiology of the disease to help the kiwifruit producers reduce losses and preserve soil health and fertility. Longitudinal studies incorporating additional environmental and host-related variables are crucial to understanding the KVDS aetiology and informing sustainable soil management practices. The impact of KVDS on microbial communities, favouring pathogenic organisms, underscores the need for strategies to restore and maintain a healthy microbial balance in the rhizosphere.

## Figures and Tables

**Figure 1 microorganisms-12-02347-f001:**
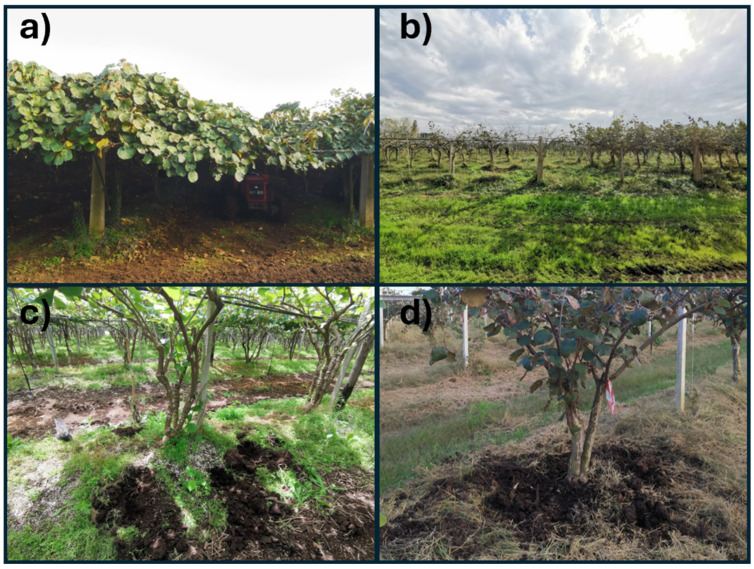
Sampling layout and orchard conditions. (**a**) Asymptomatic and (**b**) symptomatic kiwifruit orchard. Detailed view of kiwifruit trees with designated sampling points around the trunk for (**c**) asymptomatic and (**d**) symptomatic trees with the five specific sampling locations around each trunk.

**Figure 2 microorganisms-12-02347-f002:**
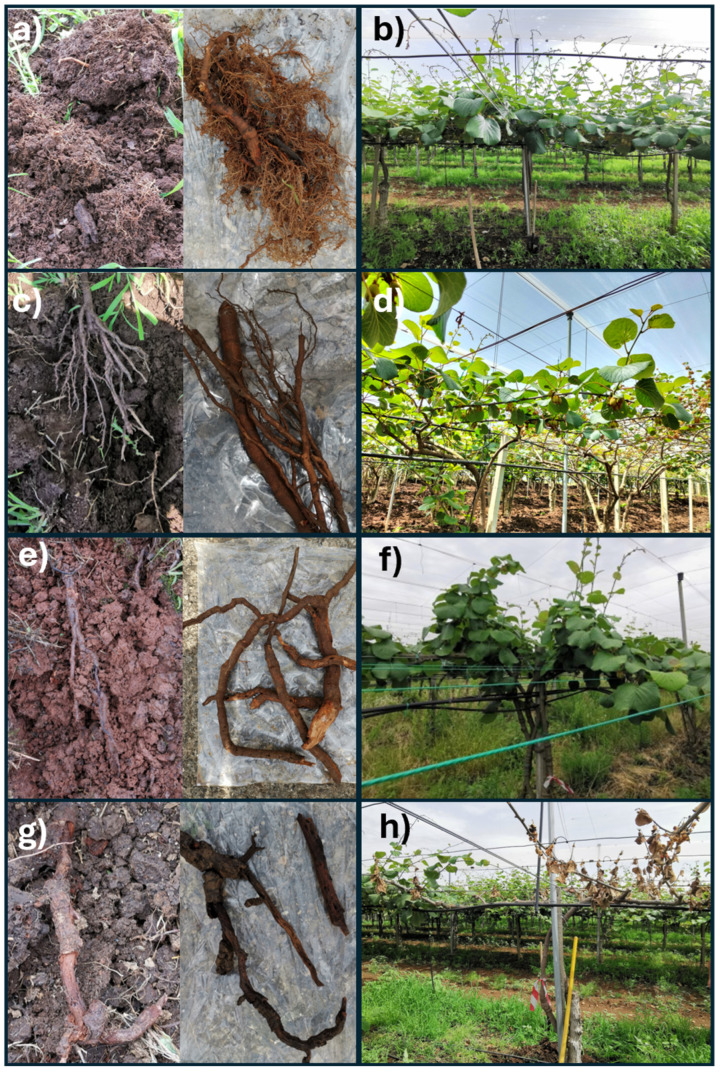
Scoring scale for disease severity (0–3) for epigeal and hypogeal symptoms in kiwifruit plants. Epigeal symptoms: (**b**) no symptoms (healthy plant); (**d**) mild symptoms (plant decline sometimes visible through reduced shoot growth, leaf chlorosis, fewer new shoots, and smaller leaves); (**f**) severe symptoms (leaf drop, fruit drop, reduced fruit size if present, and overall vine decline); (**h**) dead plant. Hypogeal symptoms: (**a**) no symptoms (healthy roots); (**c**) mild symptoms (reduction of absorbent roots and visible necrosis); (**e**) severe symptoms (decay of primary roots, near-complete rot of secondary roots, loss of cortical tissue, “rat-tail” appearance, and loss of absorbent roots); (**g**) dead roots. For root symptoms, images show roots during sampling (**left**) and after washing (**right**).

**Figure 3 microorganisms-12-02347-f003:**
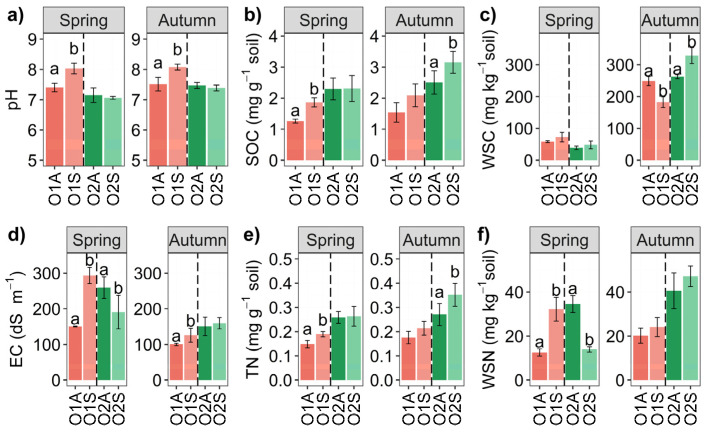
Bar chart of soil physicochemical properties of Orchards 1 and 2 in Spring and Autumn. (**a**) pH, (**b**) SOC: soil organic carbon content, (**c**) WSC: soil water-soluble C, (**d**) EC: electrical conductivity, (**e**) TN: total soil nitrogen content, (**f**) WSN: soil water-soluble N, and the letters A and S indicate Asymptomatic and Symptomatic, respectively. Different letters (a, b) indicate significant differences based on One-Way ANOVA results at *p* < 0.05. Error bars represent the standard error of the mean (4 replicates).

**Figure 4 microorganisms-12-02347-f004:**
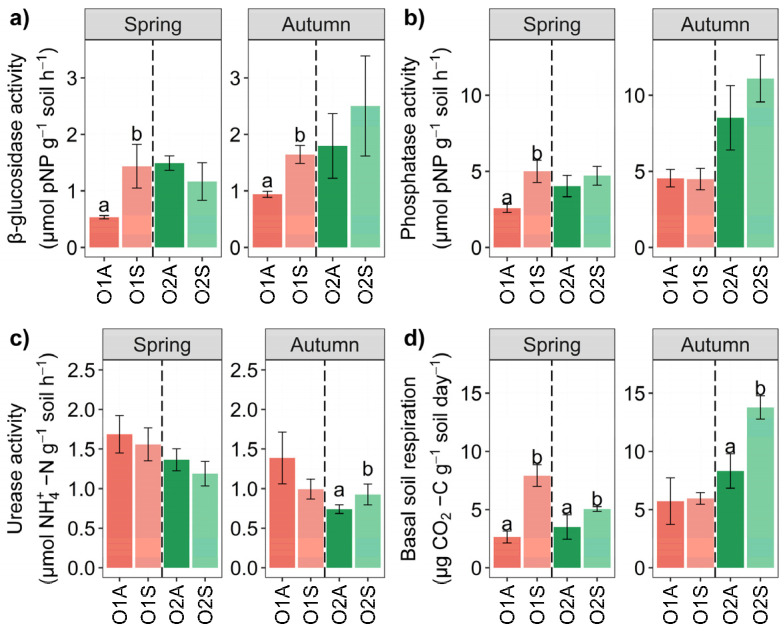
Bar chart of β-glucosidase (μmol PNF g^−1^ soil h^−1^) (**a**), alkaline phosphatase (μmol PNF g^−1^ soil h^−1^) (**b**), urease (μmol NH_4_^+^ g^−1^ soil h^−1^) (**c**), and basal soil respiration (BSR) (mg CO_2_ kg^−1^ soil day^−1^) (**d**) of Orchard 1 and 2 in Spring and Autumn. The letters A and S indicate Asymptomatic and Symptomatic, respectively. Different letters (a, b) indicate significant differences based on One-Way ANOVA results at *p* < 0.05. Error bars represent the standard error of the mean (4 replicates).

**Figure 5 microorganisms-12-02347-f005:**
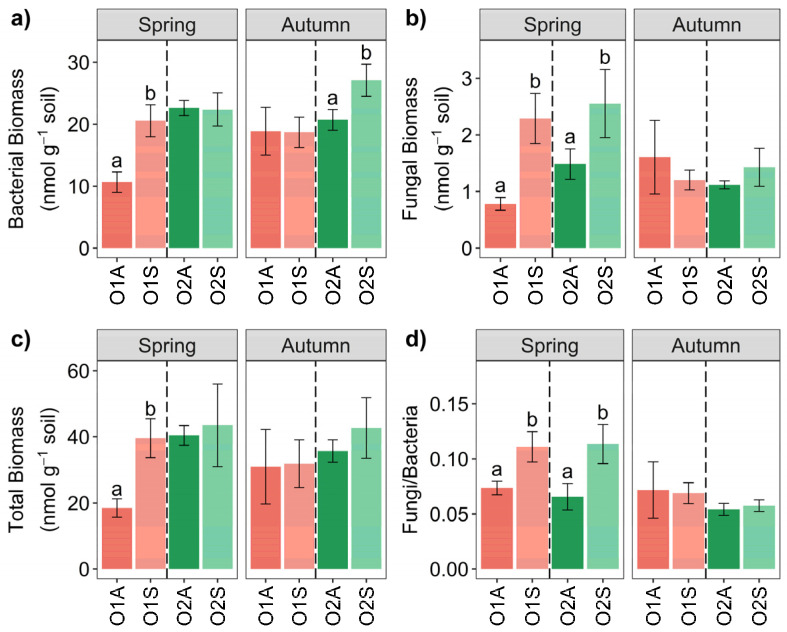
Bar chart of biomass abundance (nmol g^−1^ soil h^−1^) divided into Bacteria (**a**), Fungi (**b**), Gram- Total Biomass (**c**), Fungi/Bacteria ratio (**d**), of Orchard 1 and 2 in Spring and Autumn. Different letters (a, b) indicate significant differences based on One-Way ANOVA results at *p* < 0.05. Error bars represent the standard error of the mean (4 replicates).

**Figure 6 microorganisms-12-02347-f006:**
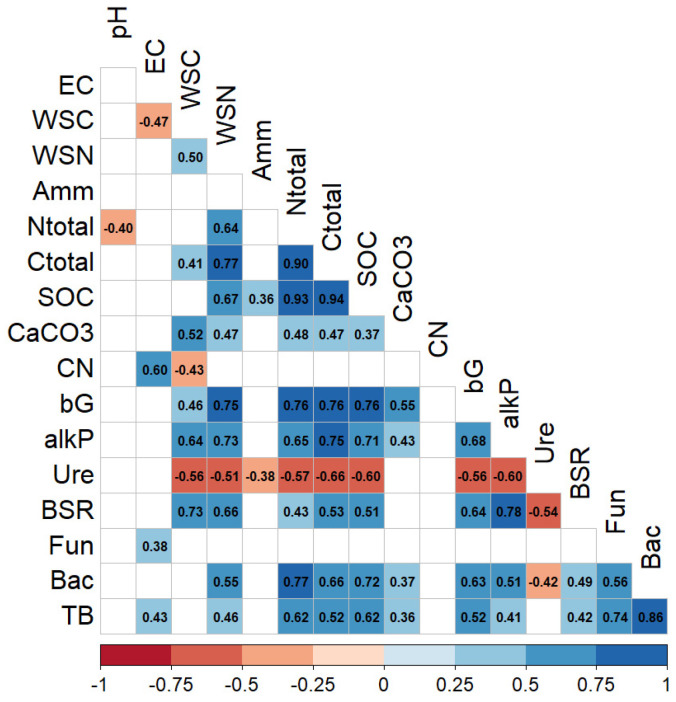
Heat map of Spearman’s correlation for soil and physiological parameters. Negative and positive correlations are represented in blue and red, respectively. All correlations are significant at *p* < 0.05. EC: electrical conductivity, WSC: soil water-soluble C, WSN: soil water-soluble N, Amm: soil ammonium content, Ntotal: total soil nitrogen content, C total: total soil carbon content, SOC: soil organic carbon, CaCO_3_: soil carbonate calcium content, CN: carbon/nitrogen ratio, bG: β-glucosidase activity, alkP: alkaline phosphatase activity, Ure: urease activity, BSR: basal soil respiration, Fun: soil fungal biomass, Bac: soil bacterial biomass, TB: total soil microbial biomass.

**Figure 7 microorganisms-12-02347-f007:**
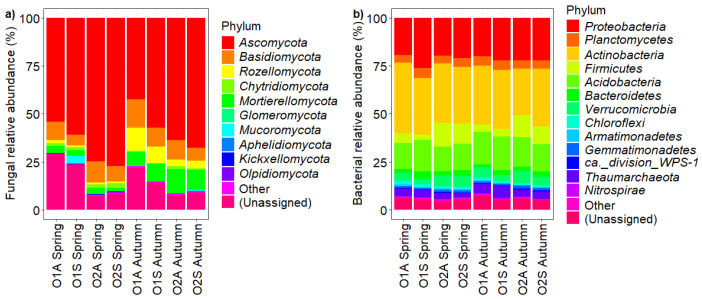
Composition of fungal (**a**) and bacterial (**b**) abundance at phylum level in Orchards 1 and 2 in Spring and Autumn.

**Figure 8 microorganisms-12-02347-f008:**
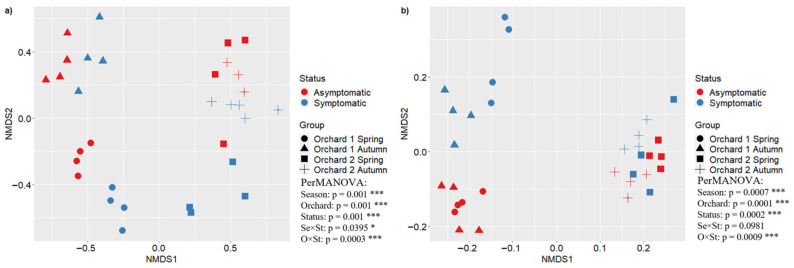
NMDS analysis of fungal community (**a**) and bacterial community (**b**) from Orchard 1 in Spring (dot) and Autumn (triangle), and Orchard 2 in Spring (square) and Autumn (cross). The asymptomatic and symptomatic samples are shown in red and blue, respectively. For the PerMANOVA test, the significance levels are shown at * *p* ≤ 0.05, and *** *p* ≤ 0.001. O×C means orchard and status interaction.

**Table 1 microorganisms-12-02347-t001:** Average scores of the epigeal and root symptoms of 4 kiwifruit plants from the 2 orchards collected in spring and autumn. O: orchards; A: Asymptomatic; S: Symptomatic.

Description		Orchard 1 (O1)		Orchard 2 (O2)	
		Asymptomatic	Symptomatic	Asymptomatic	Symptomatic
Epigeal	Spring	0	2	0	2
	Autumn	0	2	0	2
Hypogeal	Spring	0	1.75	0.25	1.25
	Autumn	0	1.75	0.75	2

**Table 2 microorganisms-12-02347-t002:** Mean value of the soil physicochemical properties, pH, electrical conductivity (dS m^−1^) EC, water-soluble C (mg kg^−1^ soil) WSC, water-soluble N (mg kg^−1^ soil) WSN, NH_4_^+^, total N, total C, soil organic carbon SOC, Calcium carbonate content CaCO_3_, total carbon/nitrogen ratio C/N, β-glucosidase (μmol PNF g^−1^ soil h^−1^), alkaline phosphatase (μmol PNF g^−1^ soil h^−1^), urease activity (μmol NH_4_^+^ g^−1^ soil h^−1^), basal soil respiration (mg CO_2_ kg^−1^ soil day^−1^) BSR, and the abundance (biomass in nmol g^−1^ soil h^−1^) of the microbial community divided into Fungi, Bacteria, Gram− and Gram+ Bacteria, Actinobacteria, Total PLFAs, Gram+/Gram− ratio, and Fungi/Bacteria ratio of Orchard 1 and 2 in Spring and Autumn. β-glucos.: β-glucosidase activity, Alk. Phos.: alkaline phosphatase activity, Urease: urease activity, BSR: basal soil respiration, Fun/Bac: Fungi/Bacteria ratio; the letters A and S indicate Asymptomatic and Symptomatic, respectively. One-way ANOVA comparison of asymptomatic and symptomatic areas for each orchard, season, and variable, with significance levels indicated by * (*p* ≤ 0.05) and ** (*p* ≤ 0.01), *** (*p* ≤ 0.01).

	Orchard 1 (O1)				Orchard 2 (O2)			
	Spring		Autumn		Spring		Autumn	
Variables	Asymptomatic	Symptomatic	Asymptomatic	Symptomatic	Asymptomatic	Symptomatic	Asymptomatic	Symptomatic
pH	7.4	8.0 ***	7.5	8.1 **	7.2	7.1	7.5	7.4
EC	150.7	293.8 ***	100.2	126.1 *	259.8	190.9 *	150.9	159.6
WSC	58.7	72.8	248.7	182.5 ***	39.3	48.3	262.3	328.6 **
WSN	12.5	32.2 ***	20.2	24.1	34.6	13.9 ***	40.6	47.2
NH_4_^+^	2.7	1.2	1.1	1.6	1.4	1.7	1.9	1.9
Total N	0.1	0.2 **	0.2	0.2	0.3	0.3	0.3	0.4 *
Total C	1.4	2.1 ***	2.0	2.4	2.5	2.3	2.9	3.5 *
SOC	1.3	1.9 ***	1.5	2.1	2.3	2.3	2.5	3.2 *
CaCO_3_	1.1	1.9	2.6	2.8	1.8	1.8	2.7	4.0
C/N	9.5	11.1 *	8.7	9.7	9.7	9.6	8.8	9.0
β-glucos.	0.54	1.44 **	0.94	1.64 ***	1.49	1.17	1.8	2.5
Alk. Phos.	2.57	5.00 ***	4.55	4.49	4.03	4.72	8.52	11.1
Urease	1.69	1.56	1.39	0.99	1.36	1.19	0.74	0.93 *
BSR	2.64	7.92 ***	5.73	5.96	3.51	5.05 *	8.32	13.78 ***
Fungi	0.8	2.3 ***	1.6	1.2	1.5	2.6 *	1.1	1.4
Bacteria	10.7	20.6 ***	18.9	18.7	22.6	22.4	20.7	27.1 **
Gram−	3.6	7.5 ***	6.7	6.5	7.4	6.7	6.6	8.9 **
Gram+	7	13.1 ***	12.2	11.7	15.2	17.4	14.1	18.0 **
Actinobac.	0.6	1.0 ***	0.7	0.6	0.9	1.0	0.5	0.7
Total PLFAs	18.5	39.6 ***	31.0	31.9	40.5	43.5	35.7	42.7
Gram+/Gram−	1.9	1.8	2.0	2.0	2.0	2.6	2.1	2.1
Fun/Bac	0.07	0.11 **	0.07	0.07	0.07	0.11 **	0.05	0.06

**Table 3 microorganisms-12-02347-t003:** Mean of Richness and Shannon index of fungal (ITS2) and bacterial (16S) communities. One-way ANOVA comparison of asymptomatic and symptomatic areas for each orchard, season, and factor, with significance levels indicated by * (*p* ≤ 0.05) and ** (*p* ≤ 0.01).

	Spring				Autumn			
Factor	O1A	O1S	O2A	O2S	O1A	O1S	O2A	O2S
ITS2 Richness	348	366	427	396	379	400	458	448
ITS2 Shannon	4.03	3.98	4.69	4.35	4.25	4.50	4.72	4.75
16S Richness	1261	1392 *	1310	1439 *	1311	1259	1354	1361
16S Shannon	6.19	6.45 **	6.24	6.41 *	6.21	6.12	6.26	6.36

**Table 4 microorganisms-12-02347-t004:** OTUs representative of symptomatic samples were obtained by Indicator Species Analysis by comparing the OTU Table of symptomatic and asymptomatic samples for both seasons. The OTU number, statistical value (Stat), *p*-value, and identification (d: domain, p: phylum, c: class, o: order, f: family, g: genera, s: species) are reported here.

OTU Number	Stat	*p*-Value	Identification
Otu258	0.506	*p* < 0.001	d: *Fungi*, p: *Ascomycota*, c: *Pezizomycetes*, o: *Pezizales*, f: *Pyronemataceae*
Otu201	0.502	*p* < 0.01	d: *Fungi*, p: *Ascomycota*, c: *Eurotiomycetes*, o: *Eurotiales*, f: *Aspergillaceae*, g: *Penicillium*
Otu806	0.490	*p* < 0.01	d: *Fungi*, p: *Ascomycota*, c: *Sordariomycetes*, o: *Hypocreales*, f: *Clavicipitaceae*, g: *Keithomyces*, s: *Keithomyces*_*indicus*
Otu27	0.486	*p* < 0.001	d: *Fungi*, p: *Ascomycota*, c: *Sordariomycetes*, o: *Sordariales*, f: *Chaetomiaceae*
Otu1041	0.484	*p* < 0.01	d: *Fungi*, p: *Ascomycota*, c: *Sordariomycetes*, o: *Hypocreales*, f: *Stachybotryaceae*, g: *Stachybotrys*, s: *Stachybotrys*_*limonisporus*
Otu69	0.479	*p* < 0.001	d: *unidentified*
Otu675	0.469	*p* < 0.01	d: *Fungi*, p: *Rozellomycota*
Otu460	0.445	*p* < 0.01	d: *unidentified*
Otu16	0.442	*p* < 0.01	d: *Fungi*, p: *Ascomycota*, c: *Sordariomycetes*, o: *Hypocreales*, f: *Nectriaceae*, g: *Fusarium*
Otu98	0.431	*p* < 0.01	d: *Fungi*, p: *Ascomycota*, c: *Dothideomycetes*, o: *Pleosporales*, f: *Massarinaceae*, g: *Stagonospora*, s: *Stagonospora*_*heteroderae*
Otu107	0.423	*p* < 0.01	d: *Fungi*, p: *Ascomycota*, c: *Eurotiomycetes*
Otu391	0.418	*p* < 0.01	d: *Fungi*, p: *Ascomycota*, c: *Pezizomycetes*, o: *Pezizales*, f: *Pyronemataceae*
Otu173	0.407	*p* < 0.01	d: *Fungi*, p: *Ascomycota*, c: *Dothideomycetes*, o: *Tubeufiales*, f: *Tubeufiaceae*, g: *Helicoma*
Otu72	0.406	*p* < 0.01	d: *Fungi*, p: *Ascomycota*, c: *Sordariomycetes*, o: *Microascales*, f: *Halosphaeriaceae*
Otu104	0.403	*p* < 0.01	d: *Fungi*, p: *Basidiomycota*, c: *Agaricomycetes*, o: *Phallales*, f: *Phallaceae*, g: *Phallus*, s: *Phallus*_*hadriani*
Otu105	0.402	*p* < 0.01	d: *Fungi*, p: *Ascomycota*, c: *Sordariomycetes*
Otu1389	0.382	*p* < 0.01	d: *Fungi*, p: *Ascomycota*, c: *Sordariomycetes*, o: *Sordariales*, f: *Chaetomiaceae*, g: *Chaetomium*
Otu591	0.355	*p* < 0.01	d: *Fungi*, p: *Ascomycota*, c: *Sordariomycetes*, o: *Coniochaetales*
Otu78	0.325	*p* < 0.001	d: *Fungi*, p: *Basidiomycota*, c: *Agaricomycetes*, o: *Phallales*, f: *Phallaceae*, g: *Phallus*
Otu274	0.299	*p* < 0.01	d: *Fungi*, p: *Ascomycota*, c: *Eurotiomycetes*, o: *Chaetothyriales*, f: *Herpotrichiellaceae*, g: *Cladophialophora*
Otu561	0.291	*p* < 0.001	d: *Fungi*, p: *Glomeromycota*, c: *Glomeromycetes*, o: *Glomerales*, f: *Glomeraceae*
Otu351	0.287	*p* < 0.01	d: *Fungi*, p: *Rozellomycota*
Otu243	0.220	*p* < 0.01	d: *Fungi*, p: *Basidiomycota*, c: *Agaricomycetes*, o: *Phallales*, f: *Clathraceae*, g: *Clathrus*, s: *Clathrus*_*ruber*

**Table 5 microorganisms-12-02347-t005:** OTUs representative of asymptomatic samples obtained by Indicator Species Analysis comparing the OTU Table of symptomatic and asymptomatic samples of both seasons. The OTU number, statistical value (Stat), *p*-value, and identification (d: domain, p: phylum, c: class, o: order, f: family, g: genera, s: species) are reported here.

OTU Number	Stat	*p*-Value	Identification
Otu258	0.506	*p* < 0.001	d: *Fungi*, p: *Ascomycota*, c: *Pezizomycetes*, o: *Pezizales*, f: *Pyronemataceae*
Otu201	0.502	*p* < 0.01	d: *Fungi*, p: *Ascomycota*, c: *Eurotiomycetes*, o: *Eurotiales*, f: *Aspergillaceae*, g: *Penicillium*
Otu806	0.490	*p* < 0.01	d: *Fungi*, p: *Ascomycota*, c: *Sordariomycetes*, o: *Hypocreales*, f: *Clavicipitaceae*, g: *Keithomyces*, s: *Keithomyces_indicus*
Otu27	0.486	*p* < 0.001	d: *Fungi*, p: *Ascomycota*, c: *Sordariomycetes*, o: *Sordariales*, f: *Chaetomiaceae*
Otu1041	0.484	*p* < 0.01	d: *Fungi*, p: *Ascomycota*, c: *Sordariomycetes*, o: *Hypocreales*, f: *Stachybotryaceae*, g: *Stachybotrys*, s: *Stachybotrys_limonisporus*
Otu69	0.479	*p* < 0.001	d: *unidentified*
Otu675	0.469	*p* < 0.01	d: *Fungi*, p: *Rozellomycota*
Otu460	0.445	*p* < 0.01	d: *unidentified*
Otu16	0.442	*p* < 0.01	d: *Fungi*, p: *Ascomycota*, c: *Sordariomycetes*, o: *Hypocreales*, f: *Nectriaceae*, g: *Fusarium*
Otu98	0.431	*p* < 0.01	d: *Fungi*, p: *Ascomycota*, c: *Dothideomycetes*, o: *Pleosporales*, f: *Massarinaceae*, g: *Stagonospora*, s: *Stagonospora_heteroderae*
Otu107	0.423	*p* < 0.01	d: *Fungi*, p: *Ascomycota*, c: *Eurotiomycetes*
Otu391	0.418	*p* < 0.01	d: *Fungi*, p: *Ascomycota*, c: *Pezizomycetes*, o: *Pezizales*, f: *Pyronemataceae*
Otu173	0.407	*p* < 0.01	d: *Fungi*, p: *Ascomycota*, c: *Dothideomycetes*, o: *Tubeufiales*, f: *Tubeufiaceae*, g: *Helicoma*
Otu72	0.406	*p* < 0.01	d: *Fungi*, p: *Ascomycota*, c: *Sordariomycetes*, o: *Microascales*, f: *Halosphaeriaceae*
Otu104	0.403	*p* < 0.01	d: *Fungi*, p: *Basidiomycota*, c: *Agaricomycetes*, o: *Phallales*, f: *Phallaceae*, g: *Phallus*, s: *Phallus_hadriani*
Otu105	0.402	*p* < 0.01	d: *Fungi*, p: *Ascomycota*, c: *Sordariomycetes*
Otu1389	0.382	*p* < 0.01	d: *Fungi*, p: *Ascomycota*, c: *Sordariomycetes*, o: *Sordariales*, f: *Chaetomiaceae*, g: *Chaetomium*
Otu591	0.355	*p* < 0.01	d: *Fungi*, p: *Ascomycota*, c: *Sordariomycetes*, o: *Coniochaetales*
Otu78	0.325	*p* < 0.001	d: *Fungi*, p: *Basidiomycota*, c: *Agaricomycetes*, o: *Phallales*, f: *Phallaceae*, g: *Phallus*
Otu274	0.299	*p* < 0.01	d: *Fungi*, p: *Ascomycota*, c: *Eurotiomycetes*, o: *Chaetothyriales*, f: *Herpotrichiellaceae*, g: *Cladophialophora*
Otu561	0.291	*p* < 0.001	d: *Fungi*, p: *Glomeromycota*, c: *Glomeromycetes*, o: *Glomerales*, f: *Glomeraceae*
Otu351	0.287	*p* < 0.01	d: *Fungi*, p: *Rozellomycota*
Otu243	0.220	*p* < 0.01	d: *Fungi*, p: *Basidiomycota*, c: *Agaricomycetes*, o: *Phallales*, f: *Clathraceae*, g: *Clathrus*, s: *Clathrus_ruber*

## Data Availability

The original contributions presented in the study are included in the article/Appendix A, further inquiries can be directed to the corresponding author.

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
