# Peer review of "Kiwifruit Vine Decline Syndrome (KVDS) Alters Soil Enzyme Activity and Microbial Community"

_microorganisms, 2024, doi:10.3390/microorganisms12112347_

Round 1

Reviewer 1 Report

Comments and Suggestions for Authors

Manuscript submitted to Microorganisms is devoted to the analysis of KVDS to the soil microbiome. Reviewer has some recommendations which may improve the paper

1) In order for the reader to be able to reproduce the results presented in the article, several photographs should be provided (I believe in the main text of the article, not in the Supplement) that will help to repeat the procedures described in sections 2.1-2.3.

2) In the Introduction section should be added information about the advisability of studying the indicators described in 2.4, 2.5.

3) Figures should be moved from the Figures section to the text of the article after mentioning the corresponding figures.

4) The captions to Figures 1, 2, 3 should clarify the meaning of the values a, b. In their current form, it is not very clear how they differ from *, **, for example, in the caption of Table 2.

5) In Table 4, spaces should follow the sign ","

6) In the Conclusions section, it would be nice to add recommendations that the authors can give to farmers and other interested parties.

Sincerely,

Comments on the Quality of English Language

I recommend that the authors submit the manuscript to a professional English-language editor

Author Response

Comment 1): In order for the reader to be able to reproduce the results presented in the article, several photographs should be provided (I believe in the main text of the article, not in the Supplement) that will help to repeat the procedures described in sections 2.1-2.3.

Response 1: Thank you for your kind suggestion. I have added a photo table with several images to illustrate the overall status of symptomatic and asymptomatic areas (Figure 1), as well as the sampling method as described in paragraph 2.1 (Figure 1). Additionally, I prepared a table detailing the typical symptoms for each epigeal and hypogeal severity class as described in paragraph 2.2. Regarding paragraph 2.3 (Figure 2). I would prefer not to include photos of the standard laboratory procedures.

Comment 2) In the Introduction section should be added information about the advisability of studying the indicators described in 2.4, 2.5.

Response 2: Thank you for the suggestion. To enhance clarity, We added lines 68-75 to the Introduction section.

Comment 3) Figures should be moved from the Figures section to the text of the article after mentioning the corresponding figures.

Response 3: I completely agree with you. Following the Microorganisms journal template, I included all figures and tables in the designated section of the file for submission.

Comment 4) The captions to Figures 1, 2, 3 should clarify the meaning of the values a, b. In their current form, it is not very clear how they differ from *, **, for example, in the caption of Table 2.

Response 4: Many thanks. Captions for Figures 1, 2, and 3 (now Figures 3, 4, and 5) have been revised on lines 337-338, 344-345, and 349-350. Captions for Tables 2 and 3 were also revised as suggested on lines 380-382 and 384-385.

Comment 5) In Table 4, spaces should follow the sign ","

Response 5: Thank you, I added the spaces in tables 4 and 5.

Comment 6) In the Conclusions section, it would be nice to add recommendations that the authors can give to farmers and other interested parties.

Response 6: In lines 487-491, in the conclusions paragraph, We added two suggestions for farmers.

Comment 7) I recommend that the authors submit the manuscript to a professional English-language editor

Response 7: Thank you for the recommendation. We have improved the manuscript readability by submitting it to CREA’s internal languageediting service. If necessary, I will submit the final version to the MDPI language editing service.

Reviewer 2 Report

Comments and Suggestions for Authors

Kiwifruit Vine Decline Syndrome (KVDS) alters soil enzyme activity and microbial community presents new information on Kiwifruit Vine Decline Syndrome (KVDS) a problem that  spread out in Italy, Relevant findings for this economic plant cultivation, specially focusing on soil health.

Authors  highlight the seasonal dynamics of soil physicochemical properties, enzyme activities, and microbial biodiversity in kiwifruit orchards affected by Kiwifruit Vine Decline Syndrome (KVDS).

they found significant differences in enzymatic activities and microbial biomass associated with KVDS symptoms. Metabarcoding distinguished microrganisms communities in symptomatic versus asymptomatic areas. 

I believe that controls may be clearly explained.

well presented paper with 6 elaborated figures and 4 tables.

do you have any image of both orchards , to show us and compare them.

Author Response

Kiwifruit Vine Decline Syndrome (KVDS) alters soil enzyme activity and microbial community presents new information on Kiwifruit Vine Decline Syndrome (KVDS) a problem that  spread out in Italy, Relevant findings for this economic plant cultivation, specially focusing on soil health.

Authors highlight the seasonal dynamics of soil physicochemical properties, enzyme activities, and microbial biodiversity in kiwifruit orchards affected by Kiwifruit Vine Decline Syndrome (KVDS).

they found significant differences in enzymatic activities and microbial biomass associated with KVDS symptoms. Metabarcoding distinguished microrganisms communities in symptomatic versus asymptomatic areas.

Comment 1) I believe that controls may be clearly explained.

Response 1: Many thanks for the suggestion. I explained that soil collected from the rhizosphere of asymptomatic kiwifruit roots has to be considered as control (paragraph 2.1 line 99). In paragraph 2.2, lines 117-118, I remarked on the utility of the scale to select the asymptomatic areas in the orchards.

Comment 2 ): well presented paper with 6 elaborated figures and 4 tables.

Response 2: Thank you very much for your positive feedback. I'm glad to hear that you found the figures and tables well-presented and elaborative. We aimed to ensure clarity and depth in illustrating our findings, so your comment is greatly appreciated.

Comment 3): do you have any image of both orchards , to show us and compare them.

Response 3: Yes, I have. I added Figures 1 and 2 with several images to illustrate the overall status of symptomatic and asymptomatic areas, as well as the sampling method described in Paragraph 2.1. Additionally, I prepared a table detailing the typical symptoms for each epigeal and hypogeal severity class, as explained in Paragraph 2.2.